# The Algorithm of Watershed Color Image Segmentation Based on Morphological Gradient

**DOI:** 10.3390/s22218202

**Published:** 2022-10-26

**Authors:** Yanyan Wu, Qian Li

**Affiliations:** College of Digital Technology and Engineering, Ningbo University of Finance and Economics, Ningbo 315175, China

**Keywords:** color image segmentation, multistage gradient, edge detection, watershed algorithm

## Abstract

The traditional watershed algorithm has the disadvantage of over-segmentation and interference with an image by reflected light. We propose an improved watershed color image segmentation algorithm. It is based on a morphological gradient. This method obtains the component gradient of a color image in a new color space is not disturbed by the reflected light. The gradient image is reconstructed by opening and closing. Therefore, the final gradient image is obtained. The maximum inter-class variance algorithm is used to obtain the threshold automatically for the final gradient image. The original gradient image is forcibly calibrated with the obtained binary labeled image, and the modified gradient image is segmented by watershed. Experimental results show that the proposed method can obtain an accurate and continuous target contour. It will achieve the minimum number of segmentation regions following human vision. Compared with similar algorithms, this way can suppress the meaningless area generated by the reflected light. It will maintain the edge information of the object well. It will improve the robustness and applicability. From the experimental results, it can be seen that compared with the region-growing method and the automatic threshold method; the proposed algorithm has a great improvement in operation efficiency, which increased by 10%. The accuracy and recall rate of the proposed algorithm is more than 0.98. Through the experimental comparison, the advantages of the proposed algorithm in object segmentation can be more intuitively illustrated.

## 1. Introduction

In the field of image content analysis and pattern recognition, image segmentation has a substantial application value. The classic problem in image processing has been studied extensively and deeply for many years. The ultimate purpose of image segmentation is to separate the target object of interest from the background for further analysis and operation. The most intuitive way that extracts the target object is to find the closed edge contour of the target. Among the many segmentation algorithms in image segmentation, the watershed algorithm is an image segmentation algorithm based on mathematical morphology and topographical and regional growth ideas. This algorithm can calculate and accurately locate the object, and can close the contour of the object. It has become a research hotspot in recent years. The proposed method can effectively solve the problem of over-segmentation of a watershed algorithm, retain the essential targets of each scale, adjust the selected parameters in the process of segmentation, and obtain a satisfactory image segmentation effect. The proposed method avoids the influence of noise on the image segmentation result. It can obtain an ideal image segmentation effect. The segmentation accuracy and efficiency are better than in other image segmentation methods. The manuscript is organized as follows. The second section briefly introduces image segmentation and watershed segmentation and briefly introduces the primary methods used in image segmentation. Section 3 describes our new method, which combines the watershed algorithm to extract the edge and then performs the *t*-test algorithm. This paper’s marking-based watershed algorithm is improved mainly in the preprocessing stage. The experimental results based on watershed segmentation are presented in Section 5 and discussed in Section 6. Finally, the conclusion is given in Section 7.

### 1.1. Image Segmentation

In recent years, image segmentation has been a challenging problem in image processing. It has also become one of the core issues in the research of image processing and recognition technology. Image segmentation [1] is the process of dividing the image into regions. Images often contain multiple objects. For example, a medical image shows various organs and tissues that are normal or diseased. In order to achieve the purpose of recognition and understanding, almost all images must be divided into regions according to specific rules. Each area represents an object (or part) to be imprinted. Its purpose is to obtain the collected image of the target or region of interest. Computer vision also has many applications in the field of engineering. In the research of tangency detection and tangency size localization of banana robot based on vision [2], the edge detection algorithm is used to segment the axis contour, and the optimal cutoff point is obtained as a scalar. In the study of seismic performance evaluation of recycled aggregate concrete steel tubular columns based on a new label-free vision method [3], a vision measurement system is developed to track the full-field deformation of specimens. The deformations measured by laser are compared with those measured by point cloud stitching and image method. All these studies have obtained good positioning accuracy.

More and more attention has been paid to the theory of mathematical morphology used in image segmentation in recent years. It has become an important research direction in the field of image segmentation. Morphology is a mathematical tool for the effective analysis of images. The research content of mathematical morphology is the geometric morphological structure of data images. Its processing process has many apparent advantages, such as good intuitiveness, concise results, and high-use efficiency. The research and application of mathematical morphology algorithms are used in image segmentation and image restoration, image enhancement, hierarchical segmentation, scale-space analysis, edge detection, morphological framework analysis, texture data detection, compression coding implementation, and many other fields.

The watershed algorithm is an image segmentation algorithm based on mathematical morphology. Because of its good practical effect, it has been one of the hot spots of image segmentation technology research in recent years. Therefore, this paper will improve and perfect the watershed algorithm. It plays its application in the field of image segmentation has a specific research significance.

### 1.2. Watershed Algorithm

Watershed Segmentation [4,5,6,7] is an image segmentation algorithm combining geomorphology and regional growth ideas. The algorithm regards a grayscale image as a topographic map, with pixel areas with a high grayscale value representing mountains and pixel areas with a low grayscale value representing low-lying areas. If it is supposed to rain, water will flow along the sides of mountains to lower places, forming “lakes”. These “lakes” in the image are called catchment basins. As the water level rises in a catchment basin, the water may spill over into other nearby catchments. If a dam is built at the junction of each catchment basin, the water will not overflow. These DAMS locations are the watershed line, the image segmentation result to be sought.

Compared with some classical edge detection algorithms (such as Sobel, Canny, and other operators), the watershed algorithm has the advantages of light burden and high computational accuracy. It takes the gradient of the image as the input and outputs the continuous edge lines with a single-pixel width. However, due to the influence of gradient noise, quantization error, and delicate texture inside the object, there may be many local “valley bottom” and “mountain peaks” in the flat area. It will form a small space after watershed transformation, leading to over-segmentation.

### 1.3. The Solution of the Problem

The algorithm in this paper converts the color image from RGB to a new color space, avoiding the interference of reflected light on the image.

The performance of image segmentation methods based on watershed transform largely depends on the algorithm used to calculate the gradient of the image to be segmented. For watersheds, the ideal gradient operator output should be equal to the input edge height, that is, the pixel gray difference between the edge’s two sides, not the edge’s slope. Ideal step edges rarely appear in natural images, usually slope edges with relatively blurred boundaries. The traditional gradient operator outputs the edge slope for this type of edge, while the morphological gradient operator can output the edge height. Because of this, the algorithm in this paper uses a multi-scale morphological gradient operator to calculate the image gradient.

Although the multi-scale morphological gradient operator is more suitable for watershed transform, its filtering feature weakens the edge information in the original image to a certain extent while removing the noise, thus shortening the gap between the edge and the noise. This will lead to a certain degree of offset or even loss of the edge contour in the bottom filling stage of the algorithm in this paper. In order to solve this problem, the weakened edges can be strengthened and modified.

After many years of research in image segmentation, people gradually realize that any single segmentation algorithm exists. It is not easy to achieve a good segmentation effect for natural images. Therefore, integrating and applying various methods has become a research hotspot in recent years. The exact locations of these edges must be found to strengthen the borders. Therefore, this paper combined the classic edge detection algorithm into the region-based watershed transform. Based on this, this paper proposes a gradient “peak enhancement” method, which can make the weakened edge contour be corrected and strengthened again.

## 2. Related Work

Mathematical morphology is an essential academic theory based on this segmentation scheme. Operated by the two most basic units of mathematical morphology: Corrosion and inflation began to expand, expounds its fundamental principle, and then from two aspects of code and simulation experiments at the same time, this paper introduces the essential operation combined by the two other morphological tools, including morphological filtering operation and opening and closing of multi-scale morphological gradient operator, the theoretical tools will play an important role in image segmentation scheme of this paper.

By adjusting the filling threshold, the number of segmented regions is controlled. At the same time, the classical canny edge detection operator is used to obtain the original color image’s edge information to strengthen and correct the edge position in the gradient map. In this way, the contour of the segmented region can be kept as close as possible to the target edge during the filling process of the bottom.

### 2.1. Background

In essence, a watershed segmentation algorithm is a kind of segmentation algorithm based on the topological theory of mathematical morphology. Its nature is to find the minimum value in the image area and then find the corresponding watershed line by simulating the immersion method. The watershed line mentioned here represents the basic contour of the target of interest in the image: the dividing line of intensity abrupt change in the picture. It obtains the regional minimum conveniently. The idea is usually reprocessed by gradient transformation. A practical solution is to limit the number of regions allowed to exist and apply prior knowledge to the over-segmentation process through preprocessing. The standard method is used to determine the number of areas. It is based on the concept of markup. A mark is a continuous flux belonging to an image that indicates the presence of a target object. The typical process of tag selection consists of two main steps: 1. Reprocessing; 2. Define a set of criteria that all tags must meet.

The forced minimum technique [8] proposed by Soille has been widely used in marker extraction of watershed segmentation by recording some marker points in gradient images. These marks are forcibly taken as the minimum value of the gradient image. The minimum value in the gradient image is shielded. Finally, the watershed operation is carried out on the modified gradient image to complete the segmentation.

The forced minimum technique can avoid over-segmentation by modifying the grayscale image so that the minimum local area appears only at the marked position. The selection of markers can range from simple processes such as linear filtering and morphological processing to complex processes involving size, shape, position, relative distance, texture content, etc. People often use prior knowledge to carry out various segmentation and more advanced tasks. Using a marker is a previous knowledge that brings the image segmentation problem. Therefore, the watershed segmentation method can provide a framework to use prior knowledge, which is an outstanding advantage. Some scholars have proposed improved marker extraction techniques, such as adaptive marker extraction [9], and multi-level marker extraction [10] The adaptive marker extraction algorithm calculates the color gradient based on the color vector. It compels the minimum depth of the gradient image and the watershed scale information to force the minimum value of the gradient image so that the image noise and dark matter texture details can be better suppressed. It can obtain the solution of watershed over-segmentation effectively. The multistage marker extraction method can improve the over-segmentation by setting the preliminary marker points and then screening the marker points again.

Many scholars have proposed improved algorithms in recent years to suppress this phenomenon of the existing watershed. Before using the watershed algorithm, the image is preprocessed to find the segmentation object more suitable for the watershed algorithm to obtain a better segmentation effect [11,12,13,14,15]. The authors of [11] proposed a target segmentation algorithm. It is based on SLIC and region growth, which achieved an ideal segmentation effect in the target segmentation of natural scene images. Laplace sharpens the image details and improves the edge hit rate of SLIC. Region growth based on the super pixel pre-segmented by SLIC can improve the algorithm’s speed and eliminate the holes in the segmentation target. In [12], H-minima technology combines fuzzy distance transform for watershed segmentation to obtain markers, but this algorithm still has some pseudo-minimum values. The authors of [13] proposed the morphological multiscale gradient to deal with step edge effectively and fuzzy edge. The authors of [14] suggests solving the two-dimensional OSTU algorithm segmentation image segmentation problem of computational complexity and noise interference. It puts forward the Lab color space is applied to the two-dimensional OSTU algorithm. First, the color image from RGB space to Lab space is combined using L channel, a channel, b channel image information for coarse segmentation. Finally, aimed at one of the links, the image information is segmented by two-dimensional OSTU. The authors of [15] proposed a watershed algorithm that combined the second-order Butterworth low-pass filter with the morphological minimum calibration technique. The markers obtained from the low-frequency part of the gradient were used to correct the original gradient image. Finally, the watershed algorithm was applied to the modified gradient image. However, these methods still cannot avoid the impact of light and edge positioning.

### 2.2. New Algorithm Description

To avoid the impact of reflected light region on color image segmentation results, the algorithm proposed in this paper first converts the image to a grayscale image. Compared with the general minima marking on the simplified illustration, this can ensure that none of the edge information is lost. Gradient images still cannot avoid that caused by the minimum of texture details and image noise, which is without meaning. It is often easy to split or divide by mistake, so a new algorithm of gradient image reconstruction using morphological structure unit opening and closing reconstruction technology is proposed. It can filter out the noise of the rebuilt gradient and texture details, and it maintains the main outline of the area of interest. Then, a threshold method of maximum inter-class variance is used to extract the markers of the target of interest. It is from the reconstructed gradient image to avoid too many false minima. Finally, the watershed algorithm is applied to the modified gradient image. The idea is to obtain a better segmentation effect.

The algorithm flow in this paper is shown in Figure 1.

### 2.3. Gradient of Morphology

Gradient images can enhance contrast. It can highlight the significant change in lights and shades in the picture. It can reflect the changing trend of gray image levels better. It needs to be emphasized that watershed segmentation is closely related to the image gradient rather than its image. Compared with the previous direct application of the watershed and the initial image segmentation, the algorithm in this paper can make the water divider obtained on the gradient image closer to the target contour in the picture.

The most significant advantage of mathematical morphology is its flexibility. The algorithm has a natural parallel implementation structure, which realizes the similar morphological analysis and processing algorithm and dramatically improves image analysis and processing speed. The nonlinear morphological gradient in the existing gradient operation methods makes the contrast degree of the extreme value in the image more obvious. It retains the relatively smooth part of the image, so image segmentation’s watershed algorithm often uses it.

Mathematical morphology comprises a set of algebraic morphology operators, and its basic operations include expansion, corrosion, opening, and closing. Various practical mathematical morphology algorithms can be derived and combined based on these operations. The function of inflation in mathematical morphology [16,17,18,19] is to incorporate the background points around the image into the object. The expansion operation will connect the two entities if the distance between the two things is relatively close. Therefore, expansion is beneficial to fill the holes in the objects after image segmentation.

As the image $F\left(x,y\right)$ by structural elements B, it is defined as the radius
(1)$$B(F)(x,y)=\mathrm{Max}\left\{F(x-u,y-v)+B(u,v)|(x-u,y-v)\in \mathrm{DF};(u,v)\in \mathrm{DB}\right\}$$

The function of corrosion in mathematical morphology is to eliminate the boundary points of objects. It can remove objects smaller than the structural elements. If there is a delicate connection between two things in which the structural elements are large enough, corrosion operation separates the two objects.

As the image $F\left(x,y\right)$ is B corrosion structure element, it is defined as
(2)$$\Theta B(F)(x,y)=\min\left\{F\left(x+u,y+v\right)-B(u,v)|(x+u,y+v\in \mathrm{DF};(u,v)\in \mathrm{DB}\right\}$$

### 2.4. Multiscale Gradient Reconstruction

In this paper, morphological open–close reconstruction operation is adopted in the reconstruction of gradient images, which is not only to remove the local extremum caused by irregular gray influence and noise in the gradient image but also to retain the extreme information of the primary contour.

Morphological reconstruction significantly transforms digital image morphology processing [20,21,22,23]. It is associated with a structural element. One represents the mark and the beginning of conversion. The remaining image represents the mask, constraining the transformation process, and a structural part defines connectivity. Geodesic expansion and geodesic corrosion are the cornerstones of mathematical morphological reconstruction. They are operated by using labeled images and mask images. They belong to the same definition domain. The mask image is usually more significant than the labeled image. Geodesic dilation is repeated dilation of the marked image using structural elements according to specific rules, preserving the region of the substantial image greater than or equal to the mask image. Geodesic etching is the repeated etching of labeled images using structural elements according to specific rules. Similarly, it preserves areas of the marked image that are less than the mask image.

Expansion and corrosion are two basic morphological operations that combine complex morphological operations, such as open and close operations. It is called available, using the same structural element to carry out the image’s etching and expansion operations. The expansion operation, followed by the corrosion operation, is called closure.

Operator turned on is “∘”. If A is turned on by B, the formula is
(3)A∘B=(AΘB)⊕B

It can be used to eliminate small objects, separate objects at thin points, and smooth the boundaries of larger objects without significantly changing their volume.

The operation of A closed by B is the result of A being expanded by B. Then it is corroded by B. If A is the original image and B is the image of the structural element, then set A is closed by the structural element B. It is denied as A•B, which is defined as
(4)A•B=(A⊕B)ΘB

It has the function of filling the small holes in the image object, connecting the adjacent thing, and smoothing its boundary without obviously changing the field and shape of the object.

### 2.5. Canny Operator

Image edge and noise are both high-frequency components in the frequency area. Simple differential extraction operation will also increase the noise in the image. In general, it should adopt the appropriate smoothing filtering before differential operation. It can reduce the impact of noise. Canny used rigorous mathematical methods to analyze this problem and deduced the optimal edge extraction operator network in the linear combination of exponential functions. The essence of the algorithm is to use a quasi-Gaussian function for smoothing operations. Then, the maximum derivative is located by the first-order differential with direction Canny operator edge detection, which is a more practical edge detection operator. It has a good performance in edge detection. Canny operator [24] edge detection uses the first derivative of the Gaussian function to achieve a good balance between noise suppression and edge detection.

## 3. Method

### 3.1. Label Extraction

Although most regional extremum and noise are eliminated in the gradient image reconstructed by morphology open–close, there are still some minimum points unrelated to the target of interest, which will cause the target of interest. It is divided into many small meaningless areas. Suppose it can obtain the points belonging to the target of interest before the watershed transformation. These points can be used to suppress the minimum issues unrelated to the target of interest. It can effectively avoid the phenomenon of over-segmentation.
(5)$$g_{r,t}^{\mathrm{bin}}(x,y)=\left\{\begin{array}{l} 1_{{\mathrm{oc}}_{B,r}^{rec}}(x,y)>t\\ 0\mathrm{otherwise} \end{array}\right\}$$

This paper uses the threshold method to extract the marks from the images reconstructed by a morphological gradient to obtain the points belonging to the target of interest. Firstly, all local minimum issues in the gradient image are detected to determine whether each minimum point is greater than the specified threshold. The facts more significant than the specified threshold are marked. After the above processing, it will obtain a binary image, and the foreground pixel of the image keeps the deep local minimum value position when the threshold is t (t is a non-negative integer), i.e.,

For each different structural element size for morphological gradient image reconstruction, it finds the deep local minimum under the specified threshold value of the binary mark image. According to type (11), it calculates the final minimum point mark image. In this way, it can combine the different sizes with the influence of the structure of elements of the picture. It ensures virtual objects that various sizes can extract and connect part of the related minima region and inhibit over-segmentation.
(6)$$g_{t}^{\mathrm{bin}}=g_{1,t}^{\mathrm{bin}}\cup g_{2,t}^{\mathrm{bin}}\cup \ldots g_{m,t}^{\mathrm{bin}}$$

M represents the maximum radius of structural elements used to reconstruct a multiscale morphological gradient. The minimum point reserved is used as the mark point to modify the gradient image. The minimum local area only appears at the mark position, and the gray value of other pixels will be pushed up to 0 as needed to delete other local minimum regions [25]. Finally, watershed transform is performed on the modified gradient image.

Even if most of the noise is removed in the gradient image reconstructed by opening and closing the morphology, some minimum points unrelated to the target subject in the picture are still not suppressed. It resulted in many meaningless areas in the segmentation results. To solve the above problems, we can use the method of tag extraction. Mark extraction marks the minimum value of the target of interest in the gradient image before applying the watershed algorithm segmentation [26,27,28,29]. It shields other redundant minimum values, only allowing the minimum value of the utilization area to be retained. The purpose of this is to minimize the occurrence of over-segmentation problems. The minimum corresponding to the gradient graph is always more profound than the minimum corresponding to the noise. This paper adopts an adaptive morphology-based extended minimum transformation technique (H-minima), which can extract the markers combined with prior knowledge. In essence, H-minima technology [30,31,32] removes local minima with a depth less than H by setting the parameter threshold value H, so selecting the threshold value is crucial to the extraction of markers. This paper uses H-minima technology to transform the gradient reconstruction image obtained in Section 3.1.

A binary labeled image $P_{\mathrm{sign}}$ can be obtained by setting the threshold value H.
(7)$$P_{\mathrm{sign}}=H_{\min}^{\left(\mathrm{Grec},H\right)}$$

It is important to note that the value of the threshold H in the traditional H-Minima technique is manually set. If the threshold value is too large, it is difficult to extract the soft edge and quickly lose the catchment basin with a small depth. If the threshold value is too small, it is challenging to eliminate intense texture noise. The exact value of the parameter can produce different segmentation effects for additional images, which results in poor robustness of the method. The process of adaptive acquisition of threshold H can be chosen to solve this problem and avoid the influence of artificial settings. This paper uses the maximum inter-class variance algorithm (minimum intra-class variance method) to obtain the threshold value. This algorithm was proposed by The Japanese scholar Zhan Yuki Otsu [33] in 1979. It is calculated based on the gray histogram and then using the least square method and is one of the best methods for automatic threshold value acquisition. The between-cluster variance algorithm principle is based on the gray histogram of the image of a target. It is the background of maximum variance between threshold selection criterion, considering the pixel area and image grayscale distribution characteristics such as whole, in after gray classification of pixels to produce ultimate conflict between groups as overall image segmentation threshold.

The set of gray image levels is set as $S=\left(1,2,3,\mathrm{and}\;I,\ldots ,L\right),$ the number of pixels with gray level I is set to $n_{i}$, then the total number of pixels of the image is
(8)$$N=n1+n2+\ldots +\mathrm{nL}=\sum \left(i\in s\right)\;\mathrm{nI}$$

After normalization, the number of pixels is P =
niN,
i∈s,
$p_{i}=1$. The gray histogram of an image is set. T is the threshold of separating two regions. According to histogram statistics of region 1, this can be separated by t. The area ratio of region 2 of the whole image and the average grayscale of the whole image, region one and region two.

The area ratio of region
1
(9)$$\theta 1=\sum \left(j=0\right)\wedge t_{\mathrm{nj}}/n$$

The area ratio of region 2
(10)$$\theta 2=\sum \left(j=t+1\right)\wedge ({G-1)}_{\mathrm{nj}}/n$$

The average gray level of the whole image
(11)$$u=\sum \left(j=0\right)\wedge (G-1)(f_{j}\times n_{j}/n)$$

The average gray of region 1
(12)u1=1θ∑(j=0)∧(G−1)(fj×nj/n)

Regional average gray level of
(13)u2=1θ∑(j=t+1)∧(G−1)(fj×nj/n)

G series for image gray.

The relation between the average gray of the whole image and the average gray value of region 1 and region 2 is
(14)$$u=u_{1}\theta_{1}+u_{2}\theta_{2}$$

The same area often has similar grey characteristics. It is characterized by the apparent gray difference between different areas. When the threshold t separation between the two areas of gray level difference is more significant, the two areas the average gray level of $u_{1}$, $u_{2}$. The whole image u is more significant. The difference between the middle gray area of variance is to describe the differences between effective parameters.

Its expression is
(15)$$\sigma B\wedge 2=\theta_{1}\times (t)\theta_{2}\times {\left(u_{1}(t)-u_{2}(t)\right)}^{2}$$

Thus, the threshold T:
T: T = Max[](16)

It is determined by the maximum variance without setting other parameters. It is a way of automatic threshold selection. It is suitable for single threshold selection in two regions and can be extended to a multi-threshold section in multiple areas.

Figure 2 Automatic thresholding method is used for image segmentation of rice, where Figure 2a the original image. Figure 2b is the automatic threshold method obtained by Figure 2a for image segmentation. Figure 2b improves the robustness and obtains a segmentation result close to the target contour.

In this study, the OTSU [33] algorithm obtains the threshold H. It is used to extract valuable markers for the gradient reconstruction image $G_{rec}$ by using the extended minimum transformation technology. The minimum position was forcibly marked to prevent the appearance of meaningless minimum and avoided the unscientific setting of threshold artificially.

### 3.2. Watershed Segmentation

The image $P_{\mathrm{sign}}$ calculates the gradient image $G_{\max}$ obtained under the new color space using the morphological minimum calibration technology [34,35,36,37,38,39]. It can correct the original local minimum value of the $G_{\max}$ image so that the minimum value retained in the modified gradient image corresponding to the area of 1 in the labeled image. $G_{\mathrm{modify}}$ represents the modified gradient image.

The operation is
(17)$$G_{\mathrm{modify}}=M_{\mathrm{or}\min}(G_{\max,}P_{\mathrm{sign}})$$

$M_{\mathrm{or}\min}()$ means the minimum value calibration operation of morphology. Then, the watershed segmentation algorithm was applied on the modified gradient image $G_{\mathrm{modify}}$. A relatively ideal segmentation effect $G_{end}$ was finally calculated. The calculation process was
(18)$$G_{\mathrm{end}}=\mathrm{WS}\left(G_{\mathrm{modify}}\right)$$

WS() represents the operator of a watershed transformation.

### 3.3. Improved Algorithm

The improved algorithm is a simulated implementation of the algorithm description, which consists of two parts: the first part is sorting, and the second part is flooding. The improved algorithm appears in the form of Algorithm 1.
**Algorithm 1** Improved algorithm**Input:**   Input image: L**Convert RGB color space to YCbCr color space, and extract three different components:**   Ib = rgb2ycbcr(I);    u11 = Ib(:,:,1);   u12 = Ib(:,:,2);   u13 = Ib(:,:,3);**Canny edge detection operator of the three components:**   uw1 = edge(u11,’canny’);    uw2 = edge(u12,’canny’);    uw3 = edge(u13,’canny’);    uw00 = uw1 | uw2 | uw3;**Used Y-component as the gray image:**   Ig = u11;**Filter the gray image on and off:**   Io = imopen(Ig,se);   Ic = imclose(Io,se);**Multiscale morphological gradient calculation:**  m1 = imdilate(Ic,b1) − imerode(Ic,b1);  m2 = imdilate(Ic,b2) − imerode(Ic,b2);  m3 = imdilate(Ic,b3) − imerode(Ic,b3);  mg1 = imerode(m1,b0);  mg2 = imerode(m2,b1);  mg3 = imerode(m3,b2);  mgf = (mg1 + mg2 + mg3)/3;**carry out bottom filling operation and the final watershed operation:**  o5 = imhmin(oo,5);   k5 = watershed(o5);  o15 = imhmin(oo,15);  k15 = watershed(o15);  o25 = imhmin(oo,25);  k25 = watershed(o25);**end**

### 3.4. Color Space Selection

The sensitive degree of brightness and color in the human visual system is a big difference [40]. In a complex image of an arbitrary point, the human eye can only identify dozens of different grayscale. Still, it can remember thousands of different colors, so in many cases, it cannot be extracted from the image using gray information alone satisfactory target. In the color images, the data used for edge detection are richer. For example, edges with the same brightness and different tones can also be detected. This useful information can be fully used in image segmentation. Color image segmentation can be regarded as applying gray image segmentation technology in various color spaces. In this section, we will briefly introduce several main color spaces.

As we all know, the color perceived by the human eye is usually called the three primary colors red (R), green (G), and blue (B). RGB is suitable for a display system but not for image segmentation and analysis because the R, G, and B components are highly relevant. As long as the brightness changes, three parts will change accordingly because the color space of RGB is very uneven, so perceptual differences between two kinds of color (the color) cannot be said for the color space. Other color feature spaces can be derived from RGB color space. However, there are many color Spaces for color image processing, no matter which one cannot replace other color spaces and is suitable for all color image processing. It is a complex problem to choose the best color space for color image segmentation.

Linear transformation space has YIQ, YUV, and YC_b_C_r_, which partially eliminate the correlation of RGB. It is a linear transformation, so the computation is small. The Y component is often used for edge detection.

The human eye’s ability to distinguish color details is far lower than brightness details. The RGB space usually represents and transforms the color image to YUV or YIQ color space. Each color space produces one luminance component signal and two chromaticity component signals. In addition, the luminance signal (Y) and chromaticity signal (U, V) are independent. The parameters used in each transformation adapt to a specific type of display equipment. YIQ or YUV space is used in color TV signals for compatibility with black and white TV and compression.

### 3.5. YUV Space

The YUV model is used in PAL TV systems, where Y stands for brightness and UV is not an abbreviation for any word. The conversion relationship between YUV and RGB is as follows:Y = 0.295R + 0.585G + 0.12B(19)
U = −0.153R − 0.317G + 0.47B(20)
V = 0.500R − 0.129G − 0.081B(21)

YUV space is equivalent to making a solution-dependent linear variation of RGB space. The ratio of U to V determines the color. (U2 + V2)1/2 represents the saturation of the color.

### 3.6. YC_b_C_r_ Space

The YIQ model is similar to the YUV model used in the NTSC TV system. The I and Q components in YIQ color space are equivalent to a rotation of UV components in YUV space, and the rotation between YIQ and RGB in relationships:Y = 0.295R + 0.585G + 0.12B(22)
I = 0.496R − 0.275G − 0.221B(23)
Q = 0.212R − 0.423G + 0.211B(24)

### 3.7. YIQ Space

YC_b_C_r_ color space is a kind of color space derived from YUV color space, which is mainly used in the digital TV system. In the conversion from RGB to YC_b_C_r_, the input and output are in 8-bit binary format, and the conversion relationship with RGB space is as follows:Y = 0.295R + 0.585G + 0.12B(25)
C_r_ = (0.500R − 0.129G − 0.081B) + 128(26)
C_b_ = (−0.153R − 0.317G + 0.47B) + 128(27)

To facilitate the application of this segmentation technology in the digital television system and to consider the compatibility of the color image and ordinary gray image simultaneously, the ITU digital television standard color space YC_b_C_r_ color space is selected as the benchmark color space of this algorithm.

### 3.8. t-Test Application

*t*-test, also known as Student’s *t*-test, is used to compare whether there is a significant difference between the mean values of two samples. One-sample *t*-test is used to test whether there is a substantial difference between the mean of a single model and a known number (the mean of a known population). Only a group of samples is tested.

The *t*-test requires that the population variance is unknown. Otherwise, the Z test (also known as the U and standard tests) can be used for average data or approximate normality.

The Matlab tool is used to extract the RGB value of the image from the edge and check whether the average value of the RGB is equal to the default requirement to test the effect of segmentation.

## 4. Experiment

In order to test the segmentation effect of the improved watershed algorithm, the experiment was completed on an Intel Core I5-8250U CPU, 8GB RAM computer, and the experiment tool was Matlab2016A. The segmentation results of PEAR, Lena, Colored Chips, Football, Scenery, and Sailing images in the figure were compared by using the region growing method. The maximum distance segmentation algorithm between OTSU classes [41] and the improved watershed algorithm in this paper.

### 4.1. Comparison of Image Segmentationbetween the Algorithm in this Paper and Other Algorithm

The content of this part mainly reflects the comparison of the algorithm and other experimental parts of the algorithm, so as to demonstrate the relevant advantages of the algorithm.

For the comparison of the three algorithms on Lena images, the experimental results are shown in Figure 3.

For the comparison of the three algorithms on Pear images, the experimental results are shown in Figure 4.

For the comparison of the three algorithms on Football images, the experimental results are shown in Figure 5.

For the comparison of the three algorithms on Colored Chips image, the experimental results are shown in Figure 6.

In order to test the validity of the data in the natural environment, the field images with mountains, rivers, and sailing were selected as the test objects. For the comparison of the three algorithms on Scenery images, the experimental results are shown in Figure 7.

For the comparison of the three algorithms on Scenery images, the experimental results are shown in Figure 8.

Given the over-segmentation phenomenon caused by the reflected light in color images, this paper selected House, Peppers, Airplane, and Lena color images to conduct experiments.

In the study, the algorithm in this paper and the method in the literature [11,15] are used to conduct experiments on the color images of Peppers. The experimental results are shown in Figure 9.

As shown in Figure 9c, there are still some pseudo-minimum values using the algorithm in [11]. Texture details are not well removed. For example, the lower part of the slender green pepper on the left is divided into two parts by a water-dividing line. In [15] algorithm, although the skinny green pepper on the left is entirely segmented, many prominent meaningless areas remain, as shown in Figure 9d. Compared with the above algorithm, the algorithm in this paper achieves a relatively good segmentation effect. It significantly reduces the light and shade noise and inhibits the influence brought by reflected light. The experimental results are shown in Figure 9b.

To facilitate the experiments and comparison, we used the image segmentation algorithms in [11,15]. The experimental results are shown in Figure 10.

### 4.2. Image Segmentation with Salt and Pepper Noise in This Paper

Salt and pepper noise is also known as impulse noise. It is produced by the image sensor, transmission channel, decoding processing black and white bright spot noise, etc. The black noise is figuratively called pepper noise. The white noise is called salt noise. Generally, the two kinds of noises appear at the same time. Image cutting often caused the white and white pepper–salt noise presented in the image. Median filtering is the most commonly used algorithm to remove pulse interference and pepper-salt noise.

Figure 11a is an image with salt and pepper noise that is added. The color image with noise added is taken as the algorithm’s input. The authors of [15] fails to segment some areas of interest in the image, resulting in over-segmentation results. The segmentation results of the algorithm in [11] show that this algorithm cannot effectively suppress the influence brought by these noises. In contrast, the algorithm in this paper can obtain straight and relatively complete object edges with good robustness, as shown in Figure 11b.

As you can see in Figure 11a, the color image of the airplane has a white reflected light surrounding the aircraft. The authors of [15] adopts RGB color space for watershed segmentation, and the experimental results are shown in Figure 11d. It can be seen that the surrounding area of the aircraft is over-segmented due to reflected light. Similarly, the segmentation results of the algorithm in [11] also have the same problem. However, the algorithm in this paper can sufficiently suppress the influence brought by the reflected light. As shown in Figure 11b, not only is the over-segmented area around the aircraft effectively removed, but the target object is completely segmented, and a good boundary effect is obtained. Compared with the above algorithm, the algorithm in this paper achieves a relatively good segmentation effect, which significantly reduces the light and shade noise and inhibits the influence brought by reflected light. The experimental results are shown in Figure 12b.

### 4.3. t-Test of the Image Segmentation Effect

The algorithm is used to extract the RGB value of the image from the edge and check whether the average value of the RGB is equal to the default requirement. The specific content of the algorithm is expressed in the form of Algorithm 2.
**Algorithm 2***t*-test of the Image Segmentation optimization**Input:**  Input Resting State**Independent Two Sample Test:**  idx = num(:,5);  x = num(:,1);  x_M = x(idx == 1);  x_F = x(idx == 0);***t* Test for homogeneity of variance:**  [p3,stats3] = vartestn(x,idx,…,’TestType’,’LeveneAbsolute’,’Display’,’off’);  disp(‘Independent *t*-test with Eyes open:’);  disp(‘Levene’s test: p = ’,num2str(p3);**if p3 < 0.05**  disp(‘Equal variances not assumed’)  [h4,p4,ci4,stats4] = test2(x_M,x_F,…,’Vartype’,’unequal’);else  disp(‘Equal variances assumed’) [h4,p4,ci4,stats4] = test2(x_M,x_F);**end**

## 5. Results

### 5.1. Comparison of Average Segmentation Time of the Algorithm

It can be seen from the comparison of experimental results in Figure 3, Figure 4, Figure 5, Figure 6, Figure 7 and Figure 8. Segmentation holes quickly appear in the process of target segmentation with the traditional region growing algorithm. It is as shown in Figure 6c. When the segmentation holes are too large, the segmented target will lose some regional information, as shown in Figure 3c and Figure 4c. Otsu segmentation in the segmentation process can grasp the integral goal of the overall outline better. It is also easy to introduce too much background, as shown in Figure 3b. The automatic gray-level threshold is 127. The gray entry of Figure 4b is 91. The difference between the target segmentation and grey values is too big. The above two kinds of algorithms will to a certain extent, lose parts of information. The automatic gray threshold obtained in Figure 5b is 121. The algorithm in this paper effectively solves the segmentation void problem existing in the traditional region-growing algorithm. It can well grasp the image’s overall contour and detailed information, as shown in Figure 5d and Figure 6d. Compared with the typical target segmentation algorithms such as OTSU threshold segmentation and region growing method, the algorithm in this paper does not introduce too much background information. As shown in Figure 3d and Figure 4d, the target region’s segmentation accuracy and detail processing have significantly improved. To intuitively evaluate the segmentation effect of the algorithm intuitively, this paper adopts the evaluation index of efficiency. The segmentation running time is shown in Table 1.

### 5.2. Comparison Table of Accuracy and Recall Rate of the Algorithm

The results in Table 1 are averaged from many experiments. It can be seen that the algorithm has a significant improvement in operating efficiency compared with the region growth method and the automatic threshold method. In Colored Chips, the region growing process takes less time than the algorithm in this paper. Because the algorithm in this paper uses multiple scales in gradient reconstruction, resulting in a certain amount of time consumption [42]. In general, the algorithm in this paper has higher efficiency than other algorithms. Experimental results show that the improved algorithm has apparent advantages in segmentation accuracy and average segmentation time over the OTSU inter-class maximum distance segmentation method and the region growing process. It has specific applicability for the segmentation of complex background images. Therefore, it is a compelling image segmentation algorithm.

This paper uses precision and recall to evaluate the three algorithms [43], as shown in Table 2. The accuracy represents the region segmented by the algorithm. The percentage of the target region is segmented. The formula is as follows:TP/(TP + FP)(28)

TP (True Positive) represents the number of pixels judged as the target region and correctly judged. FP (False Positive) represents the number of pixels regarded as the target region and wrongly assumed.

Recall rate represents the percentage of the target region to be sectioned that is correctly sectioned by the algorithm, expressed as
TP/(TP + FN)(29)

FN (False Negative) represents the number of pixels judged as non-target regions and wrongly judged. FN (False Negative) represented the number of pixels considered non-target regions and assumed wrongly.

As seen from the experimental comparison data in Table 2, recall rates of the OTSU segmentation algorithm and region growing method on the four images fluctuate around 0.9. It shows that the above two algorithms can grasp the overall contour of the segmentation target well. Still, the performance of the above two algorithms’ inaccuracy is somewhat inadequate. Especially the gray value and spatial position of the segmentation target and part of the background are similar. For example, the segmentation accuracy of the OTSU segmentation algorithm in Pears and Football images is 0.796 and 0.746. Respectively, it is unable to meet the practical application. However, the traditional region-growing algorithm is more prone to high accuracy and low recall rate. The accuracy of the conventional region growing algorithm on the four images can fluctuate around 0.9, but the relative recall rate can only reach about 0.83.

In contrast, this paper’s algorithm’s accuracy and recall rate are stable above 0.9. Especially for colored chip images, the algorithm’s accuracy and recall rate in this paper is above 0.98. We can intuitively explain the algorithm’s advantages in target segmentation through the above data comparison.

As shown in Figure 12d, a slight triangular contour is segmented on the top of the hat in the Lena image. The boundary over-segmentation is obvious. At the same time, the dividing line of the face is not apparent. However, the multiscale gradient calculation method and the maximum entropy algorithm were adopted in [15] to obtain the threshold. The second-order Butterworth low-pass filtering was used. The value of the cut-off frequency would affect the degree of removing the dark noise of the image and then lead to the change in threshold, thus affecting the segmentation effect.

As shown in Figure 12c, the black ornament on Lena’s hat is not fully segmented. Compared with the above literature algorithms, the algorithm proposed in this paper can effectively reduce the over-segmentation region and retain the edge of the central contour region. At the same time, the over-segmentation phenomenon caused by the reflected light can also be suppressed. As a result, the water division line, as shown in Figure 12b, is obtained. A complete segmentation result is finally obtained. The comparison time of these algorithms is shown in Table 3.

### 5.3. Results of t-Test of the RGB Value of the Edge

The MATLAB program runs the algorithm, and the results are shown in Table 4. The value of the *p* number is 0.07. These data indicate that the RGB edge color changes little, and the segmentation effect is perfect. This algorithm has a good segmentation effect for images with simple structure, significant contrast, and apparent edges.

## 6. Discussion

In this paper, the vital significance and application value of image segmentation are expounded, and several mainstream methods in the field of image segmentation are comprehensively and systematically analyzed: threshold segmentation method, an edge-based segmentation method, region-based edge method, and segmentation method combined with specific theoretical tools. By comparing the advantages and disadvantages of various methods, according to the specific application, reasonable selection of the Canny edge detection, mathematical morphology, and watershed algorithm, several typical segmentation theories thought put forward a complete set of color image segmentation schemes, and this kind of segmentation scheme, the simulation experiment to demonstrate the effectiveness of the scheme.

The experimental results show that the segmentation scheme in this paper achieves the following goals: (1) Through the morphological filtering of the original image and the bottom filling operation of the gradient image to a certain extent. The effects of noise and quantization error were reduced, and the over-segmentation phenomenon in Watershed’s segmentation results was apparent a significant improvement. Meaningless fragmentary blocks were significantly reduced. (2) Canny detection results are used to enhance the gradient image’s peak value to make the segmented region’s contour. The degree of fitting to the target edge is improved to a certain extent, and the segmentation result is similar to the actual target in the image. The objects are the same, and the phenomenon of missing or deviating contour is improved. (3) By adjusting the threshold of bottom filling, the number of blocks in the final segmented area is controlled. In controlling the segmentation effect, a manual intervention interface is set aside, and the segmentation depth can be manually adjusted to apply in different applications.

## 7. Conclusions

Aiming at the problem that the existing watershed algorithms still cannot solve the segmentation problem well, this paper proposes an improved labeling watershed segmentation algorithm based on new color space. Considering the segmentation effect brought by the reflected light, the algorithm in this paper converts the color image from the RGB space is transformed into the new color space. The component gradient independent of the reflected light region is obtained through calculation. The gradient image is reconstructed by the multi-scale method. The maximum interclass variance is used to obtain the threshold value automatically. It has good robustness when extracting the marker, which avoids the influence of manual identification. Many experimental results show that the proposed algorithm has good results for different segmentation requirements. The segmentation results are consistent with the target contour, which has reasonable practicability. At the same time, the algorithm’s efficiency has also been improved to a certain extent.

In the following work, we will continue with the machine learning theory. It is introduced to adjust the filling threshold adaptive, which can be free from manual supervision and fully automatic segmentation.

## Figures and Tables

**Figure 1 sensors-22-08202-f001:**
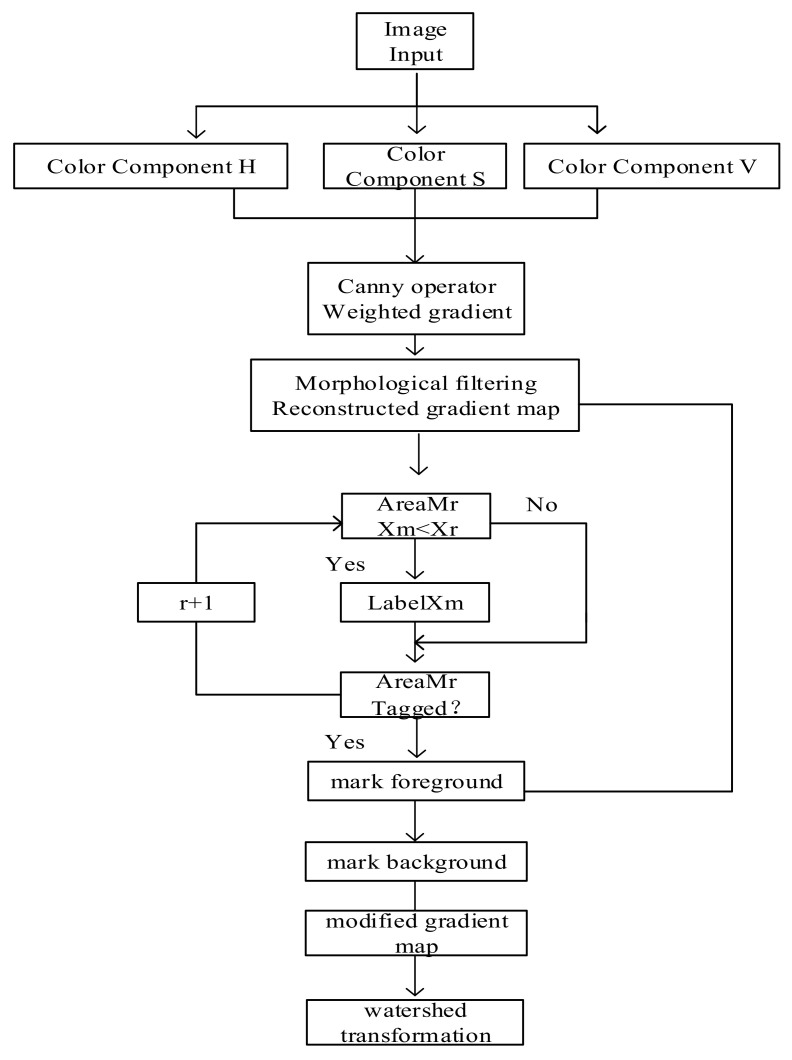
Image segmentation process of improved watershed algorithm.

**Figure 2 sensors-22-08202-f002:**
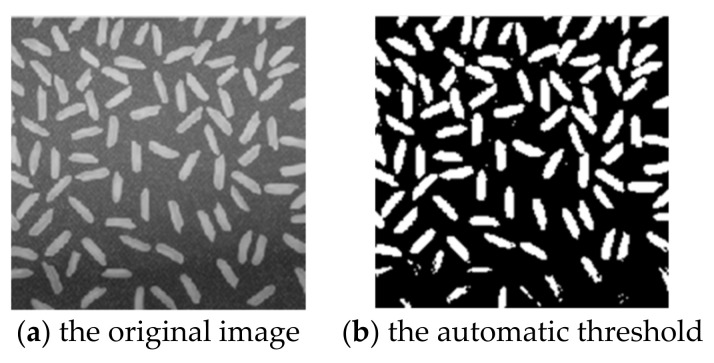
Segmentation effect of image using automatic threshold method.

**Figure 3 sensors-22-08202-f003:**
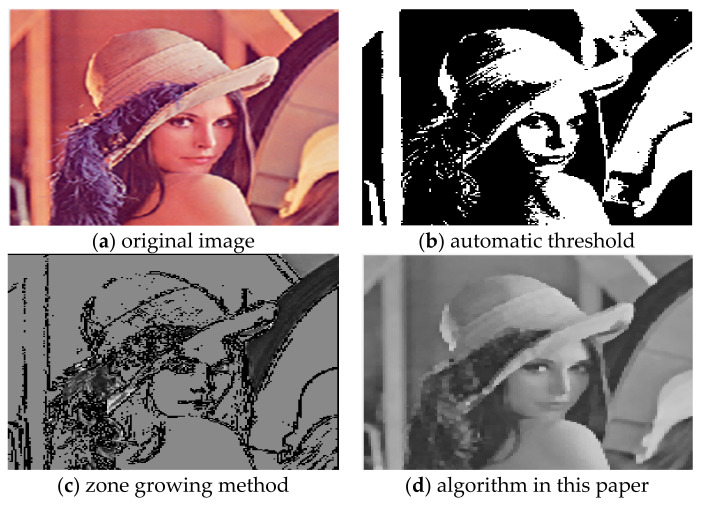
Comparison of segmentation results of Lena images.

**Figure 4 sensors-22-08202-f004:**
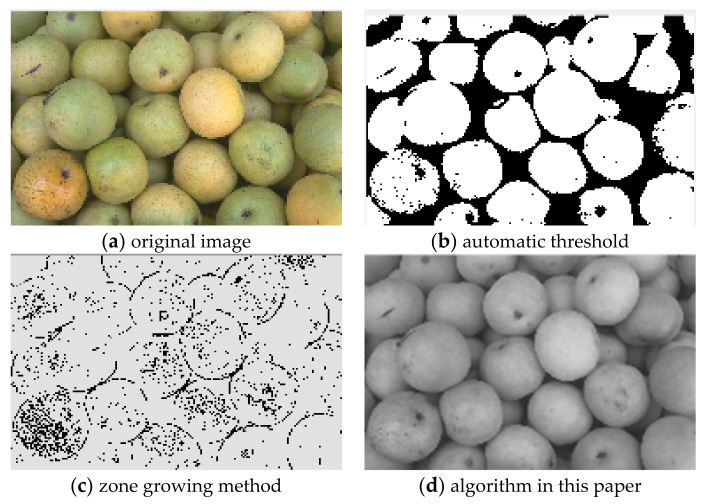
Comparison of Pear image segmentation results.

**Figure 5 sensors-22-08202-f005:**
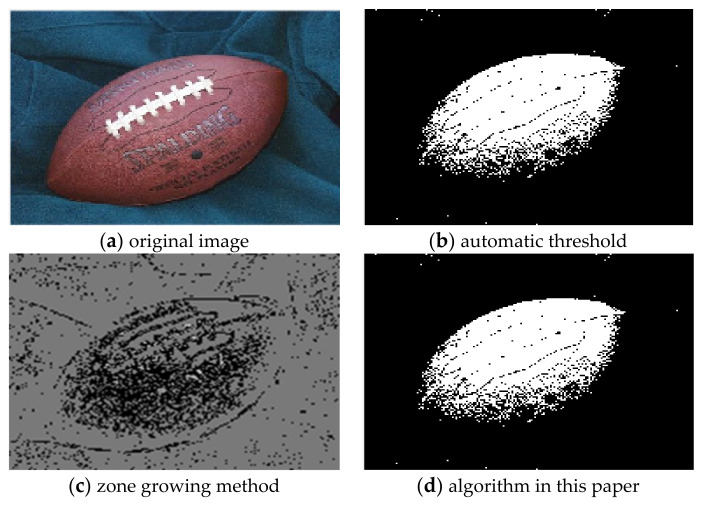
Comparison of Football image segmentation results.

**Figure 6 sensors-22-08202-f006:**
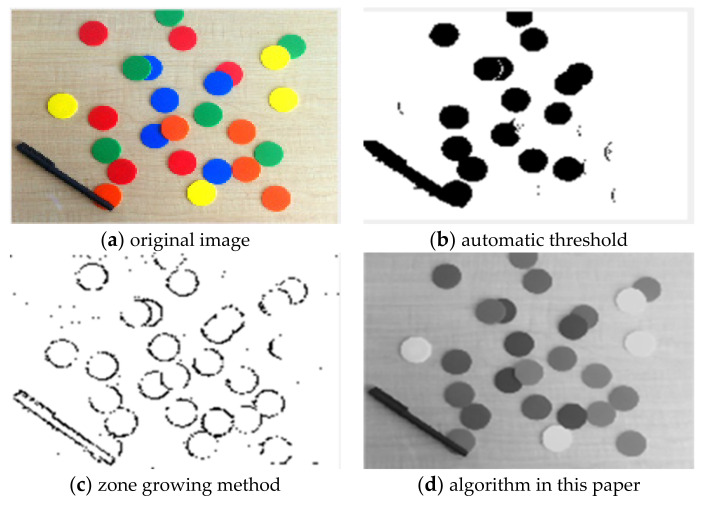
Comparison of Colored Chips image segmentation results.

**Figure 7 sensors-22-08202-f007:**
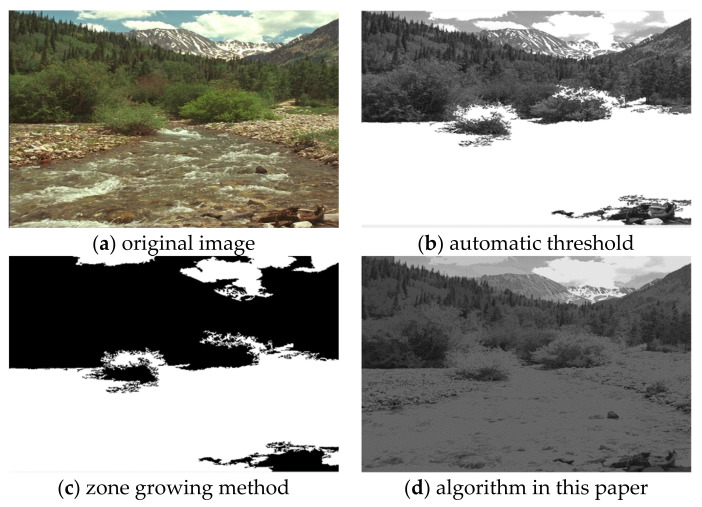
Comparison of Scenery image segmentation results.

**Figure 8 sensors-22-08202-f008:**
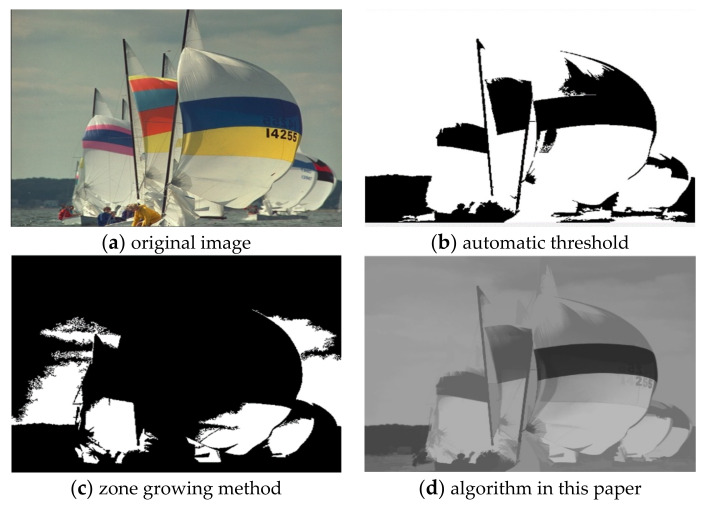
Comparison of Sailing image segmentation results.

**Figure 9 sensors-22-08202-f009:**
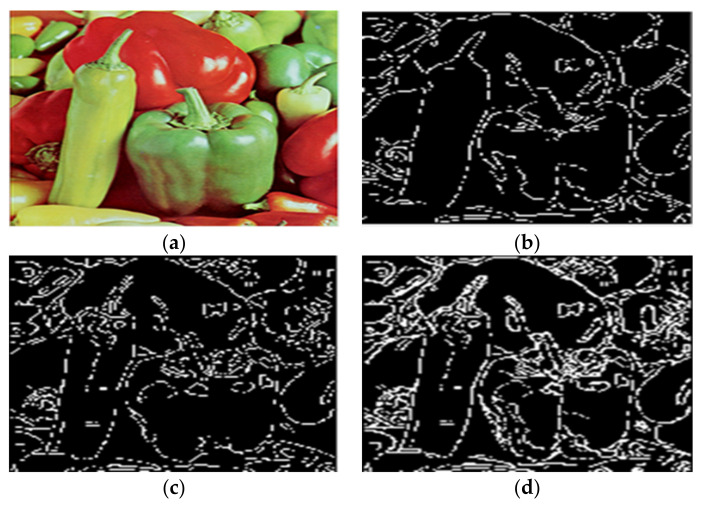
Comparison of segmentation results of Peppers by different algorithms. (**a**) The original image. (**b**) The water division lines were obtained by the algorithm in this paper. (**c**) The water division lines were obtained by the algorithm in [11]. (**d**) The water division lines were obtained by the algorithm in [15].

**Figure 10 sensors-22-08202-f010:**
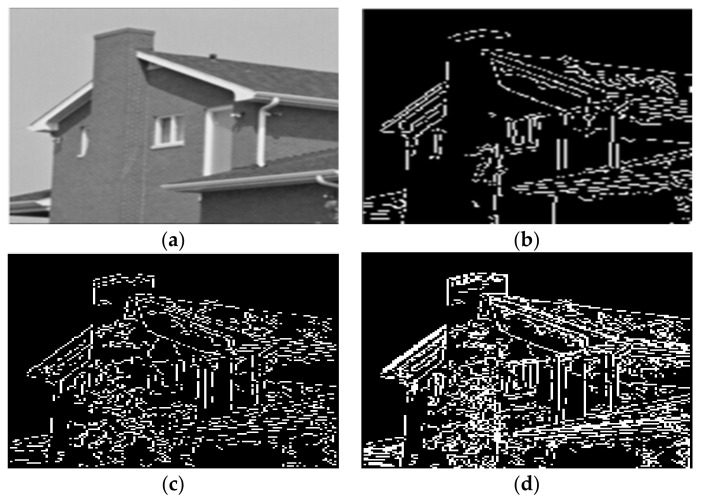
Segmentation results of House by different algorithms. (**a**) The original image. (**b**) The water division lines were obtained by the algorithm in this paper. (**c**) The water division lines were obtained by the algorithm in [11]. (**d**) The water division lines were obtained by the algorithm in [15].

**Figure 11 sensors-22-08202-f011:**
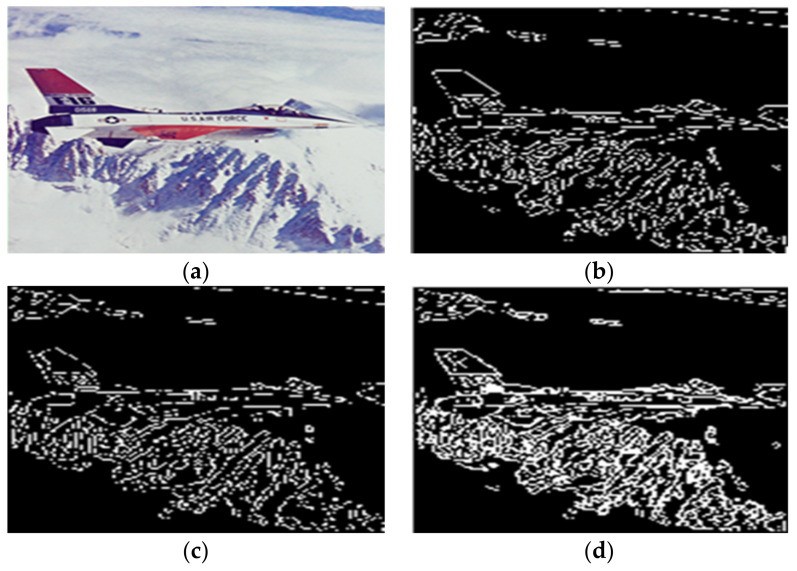
Segmentation results of Airplane by different algorithms. (**a**) The original image. (**b**) The water division lines were obtained by the algorithm in this paper. (**c**) The water division lines were obtained by the algorithm in [11]. (**d**) The water division lines were obtained by the algorithm in [15].

**Figure 12 sensors-22-08202-f012:**
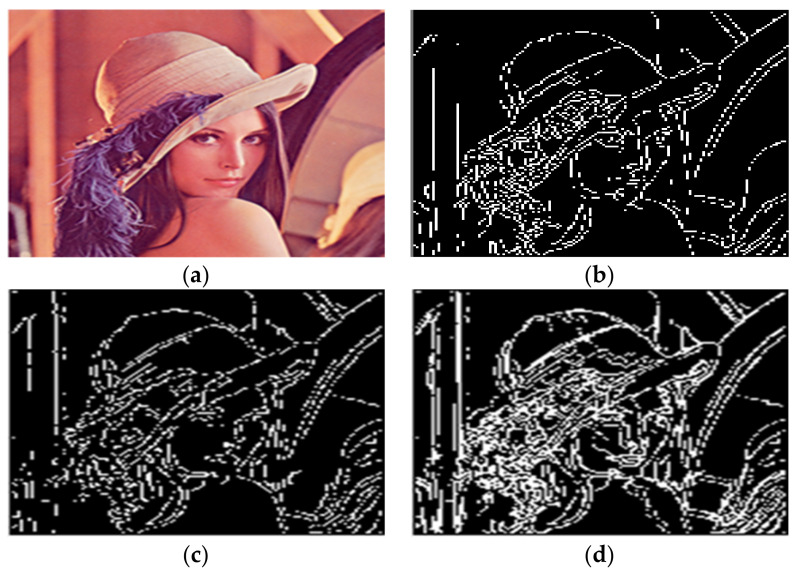
Segmentation results of LENA by different algorithms. (**a**) The original image. (**b**) The water division lines were obtained by the algorithm in this paper. (**c**) The water division lines were obtained by the algorithm in [11]. (**d**) The water division lines were obtained by the algorithm in [15].

**Table 1 sensors-22-08202-t001:** Comparison of average segmentation time of the algorithm /s.

Image	Zone Growth Method	Automatic Threshold Method	Algorithm in This Paper
Lena	0.73	3.9	2.56
Pear	2.88	2.4	2.2
Football	1.94	2.3	1.90
Colored Chips	1.69	2.5	2.19
Scenery	1.87	2.7	2.34
Sailing	1.76	2.58	2.16

**Table 2 sensors-22-08202-t002:** Comparison table of accuracy and recall rate of the algorithm.

Image	Zone Growth Method	Automatic Threshold Method	Algorithm in This Paper
Lena	Precision	0.856	0.937
	0.73		
	Recall	0.9	0.928
	0.836		
Pear	Precision	0.778	0.912
	0.796		
	Recall	0.897	0.932
	0.881		
Football	Precision	0.833	0.935
	0.746		
	Recall	0.827	0.918
	0.90		
Colored Chips	Precision	0.766	0.982
	0.735		
	Recall	0.901	0.986
	0.891		
Scenery	Precision	0.853	0.975
	0.787		
	Recall	0.867	0.898
	0.902		
Sailing	Precision	0.823	0.956
	0.727		
	Recall	0.767	0.828
	0.862		

**Table 3 sensors-22-08202-t003:** Comparison of the running time of the algorithm in this paper and the literature [11,15] for three images segmentation/s.

Image	Literature 11	Literature 15	Algorithm in This Paper
Peppers	0.736	1.952	0.938
House	0.782	1.473	0.731
Airplane	0.941	2.323	0.947
Lena	0.693	1.546	0.709

**Table 4 sensors-22-08202-t004:** Results of *t*-test of the RGB value of the edge.

Independent *t*-Test with Eyes	Open:
Levene’s test: *p* = 0.07	
Equal variances assumed	
t = −1.08	
df = 91.00	
*p* = 0.28	

## Data Availability

Not applicable.

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
