# Peer review of "The Algorithm of Watershed Color Image Segmentation Based on Morphological Gradient"

_sensors, 2022, doi:10.3390/s22218202_

Round 1

Reviewer 1 Report (New Reviewer)

This manuscript introduces an improved watershed color image segmentation algorithm based on a morphological gradient. The algorithm in this paper converts the color image from the RGB space into the new color space to avoid the interference with an image by reflected light and the experimental results provided reflect the improvement effect of the study. However, from the research background and experimental data and other aspects, the scientific value of this research could not reach the published level. There are also lots of formatting irregularities and problems caused by carelessness.

General considerations:

(1) At the thematic level, this improved algorithm solves the problem that the traditional watershed image segmentation method cannot avoid the influence of reflected light. Compared with the traditional watershed image segmentation algorithm, the proposed research is better. But from the whole development of image segmentation algorithm, the innovation of this paper is not very valuable.

(2) Technique concerns: No comparison with SOTA models based on deep learning.

Chapter 1: Introduction

(3) It is suggested that “this method” should be replaced by “the proposed method ”.

(4) In Chapter 1.1&1.2, The principles and concepts of Image Segmentation and Watershed algorithm occupy too much space, and the introduction should focus on the problems solved in this study.

Chapter 2&3: Related Work & Method

(5) Please check line 106 to 120.

(6) In Chapter 2.1, only three of the recent studies cited by the authors were published in the last five years. From the depth and breadth of research, the author 's research on image segmentation is not enough.

(7) Please edit the calculation formula according to the correct specification.

Chapter 4: Experiment

(8) The algorithms chosen by the author for comparative experiments are out of date.

(9) The amount of experimental data is insufficient. And the experimental data does not contain the data collected in the real environment.

(10) Please insert the pictures according to the correct specification.

Author Response

My response to the review is in the attached document.

Reviewer 2 Report (Previous Reviewer 1)

Thanks for your efforts. Paper can be accepted.

Round 2

Reviewer 1 Report (New Reviewer)

Clearly the authors have spent great effort in improving the work and the paper. I am happy with the quick revision but there are still few points that could be improved before the final acceptance to the work.

1. Please check line 45 to 52. There seems to be a problem with the typesetting of this paragraph.

2. Please arrange the serial numbers of all calculation formulas in a neat place.

3. Vision technology applications in various engineering fields, should also be introduced for a full glance of the scope of related areas:

Tang, Y.; Zhu, M.; Chen, Z.; Wu, C.; Chen, B.; Li, C.; Li, L. Seismic Performance Evaluation of Recycled aggregate Concrete-filled Steel tubular Columns with field strain detected via a novel mark-free vision method. Structures, 2022, 37: 426-441. https://doi.org/10.1016/j.istruc.2021.12.055

Wu, F., Duan, J., Ai, P., Chen, Z., Yang, Z., & Zou, X. Rachis detection and three-dimensional localization of cut off point for vision-based banana robot. Computers and Electronics in Agriculture,2022, 198, 107079. https://doi.org/10.1016/j.compag.2022.107079

Author Response

This manuscript is a resubmission of an earlier submission. The following is a list of the peer review reports and author responses from that submission.

Round 1

Reviewer 2 Report

The contribution of the manuscript is not significant to appear as a journal article.